# Online Stochastic Shortest Path with Bandit Feedback and Unknown Transition Function

**Aviv Rosenberg**
Tel Aviv University, Israel
`avivros007@gmail.com`

**Yishay Mansour**
Tel Aviv University, Israel
and Google Research, Israel
`mansour.yishay@gmail.com`

## Abstract

We consider online learning in episodic loop-free Markov decision processes (MDPs), where the loss function can change arbitrarily between episodes. The transition function is fixed but unknown to the learner, and the learner only observes bandit feedback (not the entire loss function). For this problem we develop no-regret algorithms that perform asymptotically as well as the best stationary policy in hindsight. Assuming that all states are reachable with probability $\beta > 0$ under any policy, we give a regret bound of $\tilde{O}(L|X|\sqrt{|A|T}/\beta)$, where $T$ is the number of episodes, $X$ is the state space, $A$ is the action space, and $L$ is the length of each episode. When this assumption is removed we give a regret bound of $\tilde{O}(L^{3/2}|X||A|^{1/4}T^{3/4})$, that holds for an arbitrary transition function. To our knowledge these are the first algorithms that in our setting handle both bandit feedback and an unknown transition function.

## 1   Introduction

Reinforcement learning is the study of problems involving sequential decision making in an unknown stochastic environment, and Markov decision process [1] is the most common model used in this field. In this model both the losses and dynamics of the environment are assumed to be stationary over time. However, in real world applications, the losses might change over time, even throughout the learning process.

To address this issue the adversarial MDP model [2] was proposed. In this model the loss function can change arbitrarily, while still assuming a fixed stochastic transition function. Since the absolute total expected loss of the learner becomes meaningless, the learner's performance is measured by the regret - comparing to the best stationary policy in hindsight.

The adversarial MDP model is actually a combination of the original MDP model with online learning [3], which considers decision making against an adversary but in a stateless environment. Online learning problems are usually divided into two types of feedback. The first is full information feedback, in which the learner observes the entire loss function after it made its decision. The second is the more challenging bandit feedback, in which the learner only observes the losses related to the actions it chose.

In this paper we propose the first algorithms for the adversarial MDP model with bandit feedback and an unknown transition function. Our algorithms are based on the recently proposed UC-O-REPS algorithm [4], that assumes unknown transition function but full information feedback. Our first algorithm, "Bounded Bandit UC-O-REPS", assumes that any state is reachable under any policy with probability $\beta > 0$ and achieves a regret bound of $\tilde{O}(L|X|\sqrt{|A|T}/\beta)$. Our second algorithm, "Shifted Bandit UC-O-REPS", removes this assumption and achieves a regret bound of $\tilde{O}(L^{3/2}|X||A|^{1/4}T^{3/4})$.

The algorithms are fairly simple, and the main challenge is the analysis of the regret and computational complexity.

## 1.1 Related Work

The works of [5] and [6] assume an unknown fixed MDP, and achieve a $\tilde{O}(L|X|\sqrt{|A|T})$ regret compared to the optimal policy. A recent work by [7] achieves $\tilde{O}(\sqrt{L|X||A|T})$ regret for large enough $T$ (which is optimal [5]), and some more recent papers achieve similar optimal results [8, 9]. We remark that the lower bound of $\Omega(\sqrt{L|X||A|T})$ by [5] also holds in our setting but might not be tight.

The work of [2], which presented the adversarial MDP model, assumes full knowledge of the transition function and full information feedback about the losses. They propose an algorithm, MDP-E, which uses an experts algorithm in each state and achieves $O(\tau^2\sqrt{T \ln |A|})$ regret, where $\tau$ is a bound on the mixing time of the MDP. Another early work in this setting, by [10], achieves an $O(T^{2/3})$ regret.

In the bandit setting, but still assuming full knowledge of the transition function, the work of [11] achieves an $O(L^2\sqrt{|A|T}/\beta)$ regret, where $\beta$ is defined similarly to our definition. Later [12] eliminate the dependence on $\beta$ but achieve only $\tilde{O}(T^{2/3})$ regret. A later work, by [13], proposed the O-REPS algorithm which guarantees an $\tilde{O}(\sqrt{L|X||A|T})$ regret.

The setting where the transition function is unknown is much more challenging and only two algorithms were previously presented for it, both assume full information feedback. The FPOP algorithm [14] achieves $\tilde{O}(L|X||A|\sqrt{T})$ regret, and the recent UC-O-REPS algorithm [4] achieves $\tilde{O}(L|X|\sqrt{|A|T})$ regret.

The rest of the paper is organized as follows. Section 2 presents the formal model and problem. Section 3 presents the concept of occupancy measures, which is the foundation of the UC-O-REPS algorithm presented in Section 4. Section 5 presents our algorithms, and sections 6 and 7 prove their regret bounds.

## 2 The Model

The Online Stochastic Shortest Path problem (OSSP) considers an episodic loop-free adversarial MDP which is defined by a tuple $M = \left(X, A, P, \{\ell_t\}_{t=1}^T\right)$, where $X$ and $A$ are the finite state and action spaces, and $P : X \times A \times X \to [0, 1]$ is the transition function such that $P(x'|x, a)$ is the probability to move to state $x'$ after performing action $a$ in state $x$.

We assume that the state space can be decomposed into $L$ non-intersecting layers $X_0, \ldots, X_L$ such that the first and the last layers are singletons, i.e., $X_0 = \{x_0\}$ and $X_L = \{x_L\}$. Furthermore, the loop-free assumption means that transitions are only possible between consecutive layers.

Let $\{\ell_t\}_{t=1}^T$ be a sequence of loss functions describing the losses at each episode, i.e., $\ell_t : X \times A \to [0, 1]$. We do not make any statistical assumption on the loss functions, i.e., they can be chosen arbitrarily.

The interaction between the learner and the environment is described in Algorithm 1. It proceeds in episodes, where in each episode the learner starts in state $x_0$ and moves forward across the consecutive layers until it reaches state $x_L$. The learner's task is to select an action at each state it visits. Alternatively, we can say that its task at each episode is to choose a stationary (stochastic) policy ,which is a mapping $\pi : X \times A \to [0, 1]$, where $\pi(a|x)$ gives the probability that action $a$ is selected in state $x$.

In episode $t$, the learner traverses a trajectory $U^{(t)} = \left(x_0^{(t)}, a_0^{(t)}, x_1^{(t)}, a_1^{(t)}, \ldots, x_{L-1}^{(t)}, a_{L-1}^{(t)}, x_L^{(t)}\right)$ using the policy $\pi_t$ it chose for this episode. That is, action $a_k^{(t)}$ is chosen using $\pi_t(\cdot|x_k^{(t)})$ and state $x_{k+1}^{(t)}$ is drawn from distribution $P(\cdot|x_k^{(t)}, a_k^{(t)})$. At the end of the episode the learner observes bandit feedback, i.e., it observes $\ell_t(U^{(t)}) = \left\{\ell_t(x_k^{(t)}, a_k^{(t)})\right\}_{k=0}^{L-1}$ and not the entire loss function $\ell_t$.

**Algorithm 1** Learner-Environment Interaction
***
**Parameters:** MDP $M = \left(X, A, P, \{\ell_t\}_{t=1}^{T}\right)$
**for** $t = 1$ **to** $T$ **do**
  learner starts in state $x_0^{(t)} = x_0$
  **for** $k = 0$ **to** $L - 1$ **do**
    learner chooses action $a_k^{(t)} \in A$
    environment draws new state $x_{k+1}^{(t)} \sim P(\cdot | x_k^{(t)}, a_k^{(t)})$
    learner observes state $x_{k+1}^{(t)}$
  learner observes suffered losses $\ell_t(x_0^{(t)}, a_0^{(t)}), \ell_t(x_1^{(t)}, a_1^{(t)}), \ldots, \ell_t(x_{L-1}^{(t)}, a_{L-1}^{(t)})$
***

For a policy $\pi$ we define its total expected loss with respect to loss function $\ell$ and transition function $P$ as

$$L(P, \pi, \ell) = \mathbb{E}\left[\sum_{k=0}^{L-1} \ell(x_k, a_k) \Big| P, \pi\right]$$

where action $a_k$ is chosen using $\pi(\cdot | x_k)$ and state $x_{k+1}$ is drawn from distribution $P(\cdot | x_k, a_k)$.

At the beginning of each episode $t$ the learner picks a policy $\pi_t$, and its goal is to minimize its total expected loss. Its performance will be measured by comparison to the best stationary policy. This is defined using the regret,

$$\hat{R}_{1:T}(P, \{\ell_t\}_{t=1}^{T}) = \sum_{t=1}^{T} L(P, \pi_t, \ell_t) - \min_{\pi} \sum_{t=1}^{T} L(P, \pi, \ell_t)$$

where the minimum is taken over all stationary stochastic policies.

## 3  Occupancy Measures

The O-REPS [13] and UC-O-REPS [4] algorithms showed that the OSSP problem can be reformulated as an online convex optimization problem, using occupancy measures on the space $X \times A \times X$. For a policy $\pi$ and a transition function $P$ the occupancy measure $q^{P,\pi}$ is defined as follows:

$$q^{P,\pi}(x, a, x') = \Pr\left[x_k = x, a_k = a, x_{k+1} = x' | P, \pi\right]$$

where $x \in X_k$ and $x' \in X_{k+1}$. We also introduce the notation $k(x)$ for the index of the layer that $x$ belongs to, and the two following notations,

$$q^{P,\pi}(x, a) = \Pr\left[x_k = x, a_k = a | P, \pi\right] = \sum_{x' \in X_{k+1}} q^{P,\pi}(x, a, x')$$

$$q^{P,\pi}(x) = \Pr\left[x_k = x | P, \pi\right] = \sum_{a \in A} q^{P,\pi}(x, a).$$

Notice that every occupancy measure $q$ induces a transition function $P^q$ and a policy $\pi^q$, that can be computed as follows:

$$P^q(x'|x, a) = \frac{q(x, a, x')}{q(x, a)} \quad ; \quad \pi^q(a|x) = \frac{q(x, a)}{q(x)}.$$

The set of all occupancy measures of an MDP $M$ is denoted as $\Delta(M)$, and can be characterized with the following lemma from [4].

**Lemma 3.1.** *For every $q \in [0, 1]^{|X| \times |A| \times |X|}$ it holds that $q \in \Delta(M)$ if and only if:*

1. $\sum_{x \in X_k} \sum_{a \in A} \sum_{x' \in X_{k+1}} q(x, a, x') = 1 \quad \forall k = 0, \ldots, L-1$

2. $\sum_{x' \in X_{k+1}} \sum_{a \in A} q(x, a, x') = \sum_{x' \in X_{k-1}} \sum_{a \in A} q(x', a, x) \quad \forall k = 1, \ldots, L-1 \forall x \in X_k$

3. $P^q = P$ *(where $P$ is the transition function of $M$)*

Thus, the task of the learner becomes selecting an occupancy measure $q^{P,\pi_t}$ at the beginning of each episode, and the regret with respect to an occupancy measures $q$ can be easily reformulated as,

$$\hat{R}_{1:T}(q, P, \{\ell_t\}_{t=1}^T) = \sum_{t=1}^T \langle q^{P,\pi_t} - q, \ell_t \rangle$$

where $\langle q, \ell \rangle \overset{def}{=} \sum_{x \in X} \sum_{a \in A} q(x,a)\ell(x,a)$. Therefore the regret of the algorithm is,

$$\hat{R}_{1:T}(P, \{\ell_t\}_{t=1}^T) = \max_{q \in \Delta(M)} \hat{R}_{1:T}(q, P, \{\ell_t\}_{t=1}^T) = \max_{q \in \Delta(M)} \sum_{t=1}^T \langle q^{P,\pi_t} - q, \ell_t \rangle.$$

## 4  The UC-O-REPS Algorithm

Our algorithms are based on the recently proposed UC-O-REPS algorithm [4]. The full details of the algorithm can be found in the original paper, but here we give a brief description.

UC-O-REPS uses the framework of UCRL2 [5] that maintains confidence sets of occupancy measures that contain $\Delta(M)$ with high probability of at least $1 - \delta$, for some parameter $0 < \delta < 1$. It proceeds in epochs such that an epoch ends every time the number of visits to some state-action pair is doubled, and the confidence set is recomputed in the beginning of every epoch. For every $(x, a, x') \in X \times A \times X_{k(x)+1}$ it keeps counters $N_i(x, a, x')$ and $N_i(x, a)$ that count the number of visits up to epoch $i$, and uses these counters to compute the empirical transition function in epoch $i$ defined as follows,

$$\bar{P}_i(x'|x,a) = \frac{N_i(x,a,x')}{\max\{1, N_i(x,a)\}}.$$

The confidence set of epoch $i$ is denoted as $\Delta(M, i)$, and it contains all the occupancy measures whose induced transition function has $L_1$-distance of at most $\epsilon_i$ from the empirical transition function, where $\epsilon_i$ is a parameter that determines the size of the confidence set and is defined as follows,

$$\epsilon_i(x,a) = \sqrt{\frac{2|X_{k(x)+1}| \ln \frac{T|X||A|}{\delta}}{\max\{1, N_i(x,a)\}}}.$$

Formally, $\Delta(M, i)$ contains all occupancy measures $q^{P',\pi}$ such that for every $(x, a)$,
$$\|P'(\cdot|x,a) - \bar{P}_i(\cdot|x,a)\|_1 \le \epsilon_i(x,a).$$

In each episode UC-O-REPS chooses an occupancy measure, from within the current confidence set, that minimizes a trade-off between the current loss function and the distance to the previously chosen occupancy measure. Formally, it performs the following steps in each episode given some parameter $\eta > 0$,

$$\tilde{q}_{t+1} = \arg\min_q \eta \langle q, \ell_t \rangle + D(q\|q_t)$$

$$q_{t+1} = q^{P_{t+1}, \pi_{t+1}} = \arg\min_{q \in \Delta(M, i(t))} D(q\|\tilde{q}_{t+1})$$

where $i(t)$ is the epoch that time step $t$ belongs to, and $D(q\|q')$ is the unnormalized KL divergence between two occupancy measures defined as $D(q\|q') = \sum_{x,a,x'} q(x,a,x') \ln \frac{q(x,a,x')}{q'(x,a,x')} - q(x,a,x') + q'(x,a,x')$.

In [4] it is shown that all the confidence sets contain $\Delta(M)$ with probability at least $1 - \delta$, that the algorithm can be implemented efficiently, and that it achieves a regret bound of $O(L|X|\sqrt{|A|T \ln T})$.

## 5  Our Algorithms

We define $\beta(M)$ as the minimum probability to visit some state under the worst exploratory policy, i.e., $\beta(M) = \min_\pi \min_{x \in X} q^{P,\pi}(x)$. Moreover, we define $p_{min}(M)$ as the minimal transition probability, that is, $p_{min}(M) = \min_{x,a,x'} P(x'|x,a)$ where $x' \in X_{k(x)+1}$.

Our first algorithm, "Bounded Bandit UC-O-REPS", is aimed for MDPs where there is a known positive lower bound on $\beta(M)$. Our second algorithm, "Shifted Bandit UC-O-REPS", works in general episodic loop-free MDPs and makes use of the first algorithm.

## 5.1 Bounded Bandit UC-O-REPS

The "Bounded Bandit UC-O-REPS" algorithm runs UC-O-REPS but with two crucial changes.

Firstly, instead of using $\ell_t$ (which we do not have) we use $\hat{\ell}_t$ which is our estimate of $\ell_t$ defined as follows,

$$\hat{\ell}_t(x,a) = \begin{cases} \frac{\ell_t(x,a)}{q_t(x,a)}, & \text{if } (x,a) \in U^{(t)} \\ 0, & \text{otherwise} \end{cases}.$$

Notice that this is a biased estimator since $P_t$ may be different from $P$,

$$
\begin{aligned}
\mathbb{E}\left[\hat{\ell}_t(x,a)\Big|U^{(1)},\dots,U^{(t-1)}\right] &= q^{P,\pi_t}(x,a)\frac{\ell_t(x,a)}{q^{P_t,\pi_t}(x,a)} \\
&= q^{P,\pi_t}(x)\pi_t(a|x)\frac{\ell_t(x,a)}{q^{P_t,\pi_t}(x)\pi_t(a|x)} \qquad (1) \\
&= q^{P,\pi_t}(x)\frac{\ell_t(x,a)}{q^{P_t,\pi_t}(x)}.
\end{aligned}
$$

Secondly, because of the bandit feedback we want to ensure that our algorithm performs enough exploration. For this purpose we constrain the confidence sets to contain only occupancy measures that visit every state with probability of at least $\alpha$, where $0 < \alpha < 1$ is a parameter. That is, we define our confidence set for epoch $i$ as $\Delta_\alpha(M,i) = \Delta(M,i) \cap \{q : q(x) \geq \alpha \quad \forall x\}$.

Thus our algorithm performs the following steps in each episode,

$$
\begin{aligned}
\tilde{q}_{t+1} &= \arg\min_q \eta\langle q, \hat{\ell}_t\rangle + D(q\|q_t) \\
q_{t+1} = q^{P_{t+1},\pi_{t+1}} &= \arg\min_{q\in\Delta_\alpha(M,i(t))} D(q\|\tilde{q}_{t+1}).
\end{aligned}
$$

If $\Delta_\alpha(M,i(t)) = \emptyset$, then $q_{t+1}$ is chosen to be an arbitrary occupancy measure. The efficient implementation of this algorithm is similar to the one of the original UC-O-REPS algorithm, and is described in details in the supplementary material (together with full pseudo-code).

## 5.2 Shifted Bandit UC-O-REPS

The "Shifted Bandit UC-O-REPS" algorithm runs "Bounded Bandit UC-O-REPS" with $\alpha = \frac{\rho}{|X|}$ (where $0 < \rho < 1$ is a parameter) but it makes the following change in order to handle the unknown $\beta(M)$ (which may be zero). It shifts the confidence sets by changing the empirical transition function. That is, instead of using $\bar{P}_i$ as the empirical transition function for epoch $i$ it uses $\bar{P}_i^\star$ which is defined as follows for every $k = 0,\dots,L-1$ and for every $(x,a,x') \in X_k \times A \times X_{k+1}$,

$$\bar{P}_i^\star(x'|x,a) = (1-\rho)\bar{P}_i(x'|x,a) + \frac{\rho}{|X_{k+1}|}.$$

To sum up, the new confidence sets are denoted as $\Delta_\alpha^\star(M,i)$ and they contain all occupancy measures $q^{P',\pi}$ such that $q^{P',\pi}(x) \geq \alpha$ for every $x$, and for every $(x,a)$,

$$\|P'(\cdot|x,a) - \bar{P}_i^\star(\cdot|x,a)\|_1 \leq \epsilon_i(x,a).$$

Clearly this algorithm can be implemented efficiently, given the efficient implementation of "Bounded Bandit UC-O-REPS" (full pseudo-code can be found in the supplementary material for completeness).

# 6 Regret Analysis - Bounded Bandit UC-O-REPS

In this case we assume that $\beta(M) > 0$ and it is known to the learner (or some positive lower bound on it). This assumption is quite strong but it holds if, for example, the minimum transition probability is not zero, i.e., $p_{min}(M) > 0$. In this case $\beta(M) \geq p_{min}(M)$.

Notice that if we run "Bounded Bandit UC-O-REPS" with $\alpha = \beta(M)$, then $\Delta(M) = \Delta_\alpha(M) \overset{def}{=} \Delta(M) \cap \{q : q(x) \geq \alpha \quad \forall x\}$. Therefore, using the proof of UC-O-REPS, we have that all the confidence sets contain $\Delta(M)$ with probability at least $1 - \delta$.

Let $q \in \Delta(M) = \Delta_\alpha(M)$, and partition the regret into two terms as follows,

$$\hat{R}_{1:T}(q, P, \{\ell_t\}_{t=1}^T) = \sum_{t=1}^T \langle q^{P,\pi_t} - q, \ell_t \rangle = \left( \sum_{t=1}^T \langle q^{P,\pi_t} - q^{P_t,\pi_t}, \ell_t \rangle \right) + \left( \sum_{t=1}^T \langle q^{P_t,\pi_t} - q, \ell_t \rangle \right)$$

The first term includes the error that comes from the estimation of the unknown transition function and will be denoted as $\hat{R}_{1:T}^{APP}$. The second term includes the error that comes from choosing sub-optimal policies and will be denoted as $\hat{R}_{1:T}^{ON}$.

Sections 6.1 and 6.2 bound these two terms and give us the following regret bound.

**Theorem 6.1.** *Let* $M = \left( X, A, P, \{\ell_t\}_{t=1}^T \right)$ *be an episodic loop-free adversarial MDP, and assume that* $\beta(M) > 0$. *Then, "Bounded Bandit UC-O-REPS" with* $\alpha = \beta(M)$ *obtains the following regret bound,*

$$\mathbb{E}\left[ \hat{R}_{1:T}(P, \{\ell_t\}_{t=1}^T) \right] \leq O\left( \frac{L|X|\sqrt{|A|T \ln T}}{\beta(M)} \right)$$

## 6.1 Bounding $\hat{R}_{1:T}^{APP}$

Recall that $\hat{R}_{1:T}^{APP}$ is the difference between the loss of the learner's chosen policies in $M$ and the loss of these policies in the "optimistic" MDPs (the ones induced by the occupancy measures $q_t$). The algorithm minimizes this difference through shrinking of the confidence sets. Notice that,

$$\hat{R}_{1:T}^{APP} = \sum_{t=1}^T \langle q^{P,\pi_t} - q^{P_t,\pi_t}, \ell_t \rangle \leq \sum_{t=1}^T \| q^{P,\pi_t} - q^{P_t,\pi_t} \|_1 \| \ell_t \|_\infty \leq \sum_{t=1}^T \| q^{P,\pi_t} - q^{P_t,\pi_t} \|_1.$$

Since the algorithm uses the same framework of confidence sets as the original UC-O-REPS (and all the confidence sets contain $\Delta(M)$ with high probability), we can use the following theorem from [4] to bound this difference.

**Theorem 6.2.** *Let* $\{\pi_t\}_{t=1}^T$ *be policies and let* $\{P_t\}_{t=1}^T$ *be transition functions such that* $q^{P_t,\pi_t} \in \Delta(M, i(t))$ *for every* $t$. *Then, when setting* $\delta = \frac{|X||A|}{T}$,

$$\mathbb{E}\left[ \hat{R}_{1:T}^{APP} \right] \leq \mathbb{E}\left[ \sum_{t=1}^T \| q^{P,\pi_t} - q^{P_t,\pi_t} \|_1 \right] \leq O\left( L|X|\sqrt{|A|T \ln T} \right)$$

## 6.2 Bounding $\hat{R}_{1:T}^{ON}$

Recall that $\hat{R}_{1:T}^{ON}$ is the regret for the performance of the online algorithm's chosen occupancy measures. Notice that the learner performs the original UC-O-REPS algorithm with respect to the sequence of loss functions $\{\hat{\ell}_t\}_{t=1}^T$ and the set of occupancy measures $\Delta_\alpha(M)$. Therefore, we can use the regret analysis of the original algorithm to obtain the following result (full proof in the supplementary material).

**Lemma 6.3.** *Let* $M = \left( X, A, P, \{\ell_t\}_{t=1}^T \right)$ *be an episodic loop-free adversarial MDP. Then, for every* $q \in \Delta_\alpha(M)$, *"Bounded Bandit UC-O-REPS" obtains,*

$$\mathbb{E}\left[ \sum_{t=1}^T \langle q^{P_t,\pi_t} - q, \hat{\ell}_t \rangle \right] \leq O\left( \frac{\eta L|A|T}{\alpha} + \frac{L \ln \frac{|X||A|}{L}}{\eta} \right)$$

Now we show that the sequence of occupancy measures chosen by the algorithm performs similarly on $\{\hat{\ell}_t\}_{t=1}^T$ and $\{\ell_t\}_{t=1}^T$ in expectation, and therefore we can derive a bound on $\hat{R}_{1:T}^{ON}$.

**Lemma 6.4.** *Let* $M = \left( X, A, P, \{\ell_t\}_{t=1}^T \right)$ *be an episodic loop-free adversarial MDP. Then, for every* $q \in \Delta_\alpha(M)$, *"Bounded Bandit UC-O-REPS" obtains,*

$$\left| \mathbb{E}\left[ \sum_{t=1}^T \langle q^{P_t,\pi_t} - q, \hat{\ell}_t \rangle \right] - \mathbb{E}\left[ \sum_{t=1}^T \langle q^{P_t,\pi_t} - q, \ell_t \rangle \right] \right| \leq O\left( \frac{L|X|\sqrt{|A|T \ln T}}{\alpha} \right)$$

*Proof.* First we use the linearity of expectation and the fact that $q_t = q^{P_t,\pi_t}$ to obtain,

$$\left| \mathbb{E}\left[\sum_{t=1}^T \langle q^{P_t,\pi_t} - q, \hat{\ell}_t\rangle\right] - \mathbb{E}\left[\sum_{t=1}^T \langle q^{P_t,\pi_t} - q, \ell_t\rangle\right] \right| = \left| \mathbb{E}\left[\sum_{t=1}^T \langle q_t - q, \hat{\ell}_t - \ell_t\rangle\right] \right|.$$

From the law of total expectation we have,

$$\left| \mathbb{E}\left[\sum_{t=1}^T \langle q_t - q, \hat{\ell}_t - \ell_t\rangle\right] \right| = \left| \mathbb{E}\left[\sum_{t=1}^T \mathbb{E}\left[\langle q_t - q, \hat{\ell}_t - \ell_t\rangle \Big| U^{(1)}, \dots, U^{(t-1)}\right]\right] \right|. \tag{2}$$

Now for every $t$ we can use the definition of $\hat{\ell}_t$ and (1) to obtain,

$$\mathbb{E}\left[\langle q_t - q, \hat{\ell}_t - \ell_t\rangle \Big| U^{(1)}, \dots, U^{(t-1)}\right] = \sum_{x,a}(q_t(x,a) - q(x,a))(q^{P,\pi_t}(x)\frac{\ell_t(x,a)}{q^{P_t,\pi_t}(x)} - \ell_t(x,a)).$$

Substituting this back into (2) we get,

$$\left| \mathbb{E}\left[\sum_{t=1}^T \langle q_t - q, \hat{\ell}_t - \ell_t\rangle\right] \right| = \left| \mathbb{E}\left[\sum_{t=1}^T \sum_{x,a}(q_t(x,a) - q(x,a))(q^{P,\pi_t}(x)\frac{\ell_t(x,a)}{q^{P_t,\pi_t}(x)} - \ell_t(x,a))\right] \right|$$

$$\leq \mathbb{E}\left[\left|\sum_{t=1}^T \sum_{x,a} \ell_t(x,a)(q_t(x,a) - q(x,a))\frac{q^{P,\pi_t}(x) - q^{P_t,\pi_t}(x)}{q^{P_t,\pi_t}(x)}\right|\right]$$

$$\leq \mathbb{E}\left[\sum_{t=1}^T \sum_x \frac{|q^{P,\pi_t}(x) - q^{P_t,\pi_t}(x)|}{q^{P_t,\pi_t}(x)}\left|\sum_a \ell_t(x,a)(q_t(x,a) - q(x,a))\right|\right]$$

$$\leq \frac{1}{\alpha}\mathbb{E}\left[\sum_{t=1}^T \sum_x |q^{P,\pi_t}(x) - q^{P_t,\pi_t}(x)|\right]$$

where the last inequality follows because $q^{P_t,\pi_t}(x) \geq \alpha$, $0 \leq \sum_a q_t(x,a)\ell_t(x,a) \leq 1$ and $0 \leq \sum_a q(x,a)\ell_t(x,a) \leq 1$. Finally, we use Theorem 6.2 to conclude that

$$\left| \mathbb{E}\left[\sum_{t=1}^T \langle q_t - q, \hat{\ell}_t - \ell_t\rangle\right] \right| \leq \frac{1}{\alpha}\mathbb{E}\left[\sum_{t=1}^T \sum_x \left|\sum_{a,x'} q^{P,\pi_t}(x,a,x') - q^{P_t,\pi_t}(x,a,x')\right|\right]$$

$$\leq \frac{1}{\alpha}\mathbb{E}\left[\sum_{t=1}^T \sum_{x,a,x'} |q^{P,\pi_t}(x,a,x') - q^{P_t,\pi_t}(x,a,x')|\right]$$

$$= \frac{1}{\alpha}\mathbb{E}\left[\sum_{t=1}^T \|q^{P,\pi_t} - q^{P_t,\pi_t}\|_1\right] \leq O\left(\frac{L|X|\sqrt{|A|T\ln T}}{\alpha}\right).$$

$\square$

**Corollary 6.1.** *Let* $M = (X, A, P, \{\ell_t\}_{t=1}^T)$ *be an episodic loop-free adversarial MDP. Then, when setting* $\eta = \sqrt{\frac{\ln \frac{|X||A|}{L}}{|A|T}}$ *and* $\delta = \frac{|X||A|}{T}$, *"Bounded Bandit UC-O-REPS" obtains,*

$$\mathbb{E}\left[\hat{R}_{1:T}^{ON}\right] \leq O\left(\frac{L|X|\sqrt{|A|T\ln T}}{\alpha}\right)$$

## 7  Regret Analysis - Shifted Bandit UC-O-REPS

We remove the assumption that $\beta(M) > 0$, and for this case will use the "Shifted Bandit UC-O-REPS" algorithm. Notice that the key insight for the regret analysis of "Bounded Bandit UC-O-REPS"

is that by setting $\alpha = \beta(M)$, we get that all the confidence sets contain $\Delta(M)$ with high probability. The idea behind "Shifted Bandit UC-O-REPS" is to work on an imaginary MDP $M^\star$ that is close to $M$ but has the property $\beta(M^\star) > 0$.

The transition function for the MDP $M^\star = \left(X, A, P^\star, \{\ell_t\}_{t=1}^T\right)$ is defined as follows for every $k = 0, \ldots, L-1$ and for every $(x, a, x') \in X_k \times A \times X_{k+1}$,

$$P^\star(x'|x, a) = (1 - \rho)P(x'|x, a) + \frac{\rho}{|X_{k+1}|}.$$

This means that the minimal transition probability is positive, i.e., $p_{min}(M^\star) \geq \frac{\rho}{|X|} > 0$. Therefore, $\beta(M^\star) \geq \frac{\rho}{|X|} > 0$ and we can run "Bounded Bandit UC-O-REPS" on $M^\star$. The problem is that our data is sampled from $M$, but we need to build confidence sets that contain $\Delta(M^\star)$ and not $\Delta(M)$. The following lemma shows that shifting the confidence sets obtains this desired property: all the confidence sets contain $\Delta(M^\star)$ with probability at least $1 - \delta$.

**Lemma 7.1.** *If $\Delta(M) \subseteq \Delta(M, i)$, then $\Delta(M^\star) \subseteq \Delta_\alpha^\star(M, i)$.*

*Proof.* Let $q^{P^\star, \pi} \in \Delta(M^\star)$. First of all, since $\beta(M^\star) \geq \frac{\rho}{|X|} = \alpha$ we have that $q^{P^\star, \pi}(x) \geq \alpha$ for every $x$. Now, Since $\Delta(M) \subseteq \Delta(M, i)$ we have that for every $(x, a)$,

$$\|\bar{P}_i(\cdot|x, a) - P(\cdot|x, a)\|_1 \leq \epsilon_i(x, a).$$

By the definition of $\bar{P}_i^\star$ and $P^\star$ we have that,

$$\|\bar{P}_i^\star(\cdot|x, a) - P^\star(\cdot|x, a)\|_1 = \sum_{x'} |\bar{P}_i^\star(x'|x, a) - P^\star(x'|x, a)|$$

$$= \sum_{x'} \left|(1 - \rho)\bar{P}_i(x'|x, a) + \frac{\rho}{|X_{k+1}|} - (1 - \rho)P(x'|x, a) - \frac{\rho}{|X_{k+1}|}\right|$$

$$= (1 - \rho) \sum_{x'} |\bar{P}_i(x'|x, a) - P(x'|x, a)|$$

$$= (1 - \rho)\|\bar{P}_i(\cdot|x, a) - P(\cdot|x, a)\|_1 \leq \epsilon_i(x, a)$$

and therefore $q^{P^\star, \pi} \in \Delta_\alpha^\star(M, i)$ and $\Delta(M^\star) \subseteq \Delta_\alpha^\star(M, i)$. $\qquad\square$

Now we divide the regret into two parts: the regret of "Bounded Bandit UC-O-REPS" in $M^\star$ and the difference in the performance of policies in $M$ and $M^\star$. Formally, the regret of any $q = q^{P, \pi} \in \Delta(M)$ is partitioned as follows,

$$\hat{R}_{1:T}(q^{P, \pi}, P, \{\ell_t\}_{t=1}^T) = \sum_{t=1}^T \langle q^{P, \pi_t} - q^{P, \pi}, \ell_t \rangle$$

$$= \left(\sum_{t=1}^T \langle q^{P, \pi_t} - q^{P^\star, \pi_t}, \ell_t \rangle\right) + \left(\sum_{t=1}^T \langle q^{P^\star, \pi_t} - q^{P^\star, \pi}, \ell_t \rangle\right) + \left(\sum_{t=1}^T \langle q^{P^\star, \pi} - q^{P, \pi}, \ell_t \rangle\right).$$

Since $\|P(\cdot|x, a) - P^\star(\cdot|x, a)\|_1 \leq 2\rho$ for every $(x, a)$, we can use Corollary E.2 in the supplementary material to bound the first and third terms as $O(\rho L^2 T)$. The second term includes the regret of "Bounded Bandit UC-O-REPS" in $M^\star$ so according to Theorem 6.1 we can bound it as $O\left(\frac{L|X|\sqrt{|A|T \ln T}}{\rho/|X|}\right) = O\left(\frac{L|X|^2\sqrt{|A|T \ln T}}{\rho}\right)$. Thus we get the following regret bound.

**Theorem 7.2.** *Let $M = \left(X, A, P, \{\ell_t\}_{t=1}^T\right)$ be an episodic loop-free adversarial MDP. Then, "Shifted Bandit UC-O-REPS" with $\rho = \frac{|X|}{\sqrt{L}}\sqrt[4]{\frac{|A| \ln T}{T}}$ obtains the following regret bound,*

$$\mathbb{E}\left[\hat{R}_{1:T}(P, \{\ell_t\}_{t=1}^T)\right] \leq O\left(L^{3/2}|X||A|^{1/4}T^{3/4} \ln^{1/4} T\right)$$

**Remark 7.1.** *One might wonder if a regret of $O(T^{2/3})$ is achievable by an algorithm that first explores to estimate the transition function and then runs a known algorithm that assumes full knowledge of the transition function. While this approach is possible in the classic online learning setting, it is unclear how to implement it in our setting since estimating the transition function properly should take about $T^{2/3}/\beta(M)$ episodes. This becomes even more complicated when $\beta(M) = 0$.*

# 8 Conclusions and Future Work

In this paper we considered online learning in episodic loop-free adversarial MDPs where the losses can change arbitrarily between episodes. We assumed the transition function is completely unknown to the learner and that it only observes bandit feedback. Our algorithms are based on the recently proposed UC-O-REPS algorithm and achieve $\tilde{O}(L^{3/2}|X||A|^{1/4}T^{3/4})$ regret for the general case, and $\tilde{O}(L|X|\sqrt{|A|T}/\beta)$ regret for the case where any state is reachable under any policy with probability at least $\beta > 0$.

Throughout the paper we assumed that the MDP is loop-free. However, it is important to mention that any episodic MDP can be transformed into a loop-free MDP by duplicating the state space $L$ times, i.e., a state $x$ becomes states $(x, k)$ where $k = 0, \ldots, L$. Therefore, our algorithms work in any episodic MDP but the regret bound has a worse dependence on $L$.

The algorithms we proposed are the first to handle the setting of both an unknown transition function and a bandit feedback, but our results are still far from the known lower bound of $\Omega(\sqrt{L|X||A|T})$. Future work should be done to find algorithms with stronger regret bounds and specifically ones that remove the loop-free assumption and still get good dependence on $L$.

Another line of work to be considered is finding tighter lower bounds for this specific problem. Since this problem is much more difficult than the usual reinforcement learning setting and techniques that involve the value function cannot be implemented here naturally, it might be the case that the lower bound of [5] cannot be achieved in this setting.

## Acknowledgements

This work was supported in part by a grant from the Israel Science Foundation (ISF) and by the Tel Aviv University Yandex Initiative in Machine Learning.

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
