[Supplementary Material · supp.pdf]

# A    efficient implementation of "Bounded Bandit UC-O-REPS"

As described in the paper, the algorithm needs to perform the following steps in each episode (while maintaining confidence sets throughout the algorithm),

$$\tilde{q}_{t+1} = \arg\min_q \eta\langle q, \hat{\ell}_t\rangle + D(q\|q_t) \tag{1}$$

$$q_{t+1} = \arg\min_{q\in\Delta_\alpha(M,i(t))} D(q\|\tilde{q}_{t+1}). \tag{2}$$

The confidence sets are maintained like in the original UC-O-REPS algorithm, and step (1) can be easily solved by setting $\tilde{q}_{t+1}(x,a,x') = q_t(x,a,x')e^{-\eta\hat{\ell}_t(x,a)}$ for every $(x,a,x') \in X \times A \times X_{k(x)+1}$. Thus, we need to describe an efficient implementation to step (2). This step can be reformulated as the following constrained convex optimization problem:

$$\min_{q,\epsilon} D(q\|\tilde{q}_{t+1})$$

$$s.t. \sum_{x\in X_k}\sum_{a\in A}\sum_{x'\in X_{k+1}} q(x,a,x') = 1 \qquad\qquad \forall k = 0,\dots,L-1$$

$$\sum_{x'\in X_{k+1}}\sum_{a\in A} q(x,a,x') = \sum_{x'\in X_{k-1}}\sum_{a\in A} q(x',a,x) \qquad\qquad \forall k = 1,\dots,L-1 \quad \forall x \in X_k$$

$$q(x,a,x') - \bar{P}_{i(t)}(x'|x,a)\sum_{y\in X_{k+1}} q(x,a,y) \le \epsilon(x,a,x') \quad \forall k = 0,\dots,L-1 \quad \forall(x,a,x') \in X_k \times A \times X_{k+1}$$

$$\bar{P}_{i(t)}(x'|x,a)\sum_{y\in X_{k+1}} q(x,a,y) - q(x,a,x') \le \epsilon(x,a,x') \quad \forall k = 0,\dots,L-1 \quad \forall(x,a,x') \in X_k \times A \times X_{k+1}$$

$$\sum_{x'\in X_{k+1}} \epsilon(x,a,x') \le \epsilon_{i(t)}(x,a)\sum_{x'\in X_{k+1}} q(x,a,x') \qquad\qquad \forall k = 0,\dots,L-1 \quad \forall(x,a) \in X_k \times A$$

$$\sum_{x'\in X_{k+1}}\sum_{a\in A} q(x,a,x') \ge \alpha \qquad\qquad \forall k = 1,\dots,L-1 \quad \forall x \in X_k$$

$$q(x,a,x') \ge 0 \qquad\qquad \forall k = 0,\dots,L-1 \quad \forall(x,a,x') \in X_k \times A \times X_{k+1}$$

An explanation to this formulation can be found in [4].

Now we will derive the solution to this problem using Lagrange multipliers. First we write the Lagrangian with $\lambda, \beta, \mu, \mu^+, \mu^-, c$ as Lagrange multipliers. Notice that we omit the non-negativity constraints, which we

can justify since the solution will be non-negative anyway.

$$\mathcal{L}(q,\epsilon) = D(q||\tilde{q}_{t+1}) + \sum_{k=0}^{L-1} \lambda_k \left( \sum_{x \in X_k} \sum_{a \in A} \sum_{x' \in X_{k+1}} q(x,a,x') - 1 \right)$$

$$+ \sum_{k=1}^{L-1} \sum_{x \in X_k} \beta(x) \left( \sum_{a \in A} \sum_{x' \in X_{k+1}} q(x,a,x') - \sum_{a \in A} \sum_{x' \in X_{k-1}} q(x',a,x) \right)$$

$$+ \sum_{k=0}^{L-1} \sum_{x \in X_k} \sum_{a \in A} \sum_{x' \in X_{k+1}} \mu^+(x,a,x') \left( q(x,a,x') - \bar{P}_{i(t)}(x'|x,a) \sum_{y \in X_{k+1}} q(x,a,y) - \epsilon(x,a,x') \right)$$

$$+ \sum_{k=0}^{L-1} \sum_{x \in X_k} \sum_{a \in A} \sum_{x' \in X_{k+1}} \mu^-(x,a,x') \left( \bar{P}_{i(t)}(x'|x,a) \sum_{y \in X_{k+1}} q(x,a,y) - q(x,a,x') - \epsilon(x,a,x') \right)$$

$$+ \sum_{k=0}^{L-1} \sum_{x \in X_k} \sum_{a \in A} \mu(x,a) \left( \sum_{x' \in X_{k+1}} \epsilon(x,a,x') - \epsilon_{i(t)}(x,a) \sum_{x' \in X_{k+1}} q(x,a,x') \right)$$

$$+ \sum_{k=0}^{L-1} \sum_{x \in X_k} c(x) \left( \alpha - \sum_{x' \in X_{k+1}} \sum_{a \in A} q(x,a,x') \right)$$

Let $(x,a,x') \in X \times A \times X_{k(x)+1}$ and consider the derivative with respect to $\epsilon(x,a,x')$.

$$\frac{\partial \mathcal{L}}{\partial \epsilon(x,a,x')} = -\mu^+(x,a,x') - \mu^-(x,a,x') + \mu(x,a)$$

So setting the gradient to zero we obtain

$$\mu(x,a) = \mu^+(x,a,x') + \mu^-(x,a,x')$$

Thus, we can discard $\mu(x,a)$ to obtain an equivalent Lagrangian. Notice that this way we also get rid of the $\epsilon(x,a,x')$ variables.

$$\mathcal{L}(q) = D(q||\tilde{q}_{t+1}) + \sum_{k=0}^{L-1} \lambda_k \left( \sum_{x \in X_k} \sum_{a \in A} \sum_{x' \in X_{k+1}} q(x,a,x') - 1 \right)$$

$$+ \sum_{k=1}^{L-1} \sum_{x \in X_k} \beta(x) \left( \sum_{a \in A} \sum_{x' \in X_{k+1}} q(x,a,x') - \sum_{a \in A} \sum_{x' \in X_{k-1}} q(x',a,x) \right)$$

$$+ \sum_{k=0}^{L-1} \sum_{x \in X_k} \sum_{a \in A} \sum_{x' \in X_{k+1}} \mu^+(x,a,x') \left( (1 - \epsilon_{i(t)}(x,a))q(x,a,x') - \bar{P}_{i(t)}(x'|x,a) \sum_{y \in X_{k+1}} q(x,a,y) \right)$$

$$+ \sum_{k=0}^{L-1} \sum_{x \in X_k} \sum_{a \in A} \sum_{x' \in X_{k+1}} \mu^-(x,a,x') \left( \bar{P}_{i(t)}(x'|x,a) \sum_{y \in X_{k+1}} q(x,a,y) - (1 + \epsilon_{i(t)}(x,a))q(x,a,x') \right)$$

$$+ \sum_{k=0}^{L-1} \sum_{x \in X_k} c(x) \left( \alpha - \sum_{x' \in X_{k+1}} \sum_{a \in A} q(x,a,x') \right)$$

Now we consider the derivative with respect to $q(x, a, x')$. We denote $\beta(x_0) = \beta(x_L) = c(x_0) = c(x_L) = 0$ to avoid addressing the edge cases explicitly.

$$\frac{\partial \mathcal{L}}{\partial q(x, a, x')} = \ln q(x, a, x') - \ln \tilde{q}_{t+1}(x, a, x') + \lambda_k + \beta(x) - \beta(x') - c(x)$$
$$+ (1 - \epsilon_{i(t)}(x, a))\mu^+(x, a, x') - (1 + \epsilon_{i(t)}(x, a))\mu^-(x, a, x')$$
$$+ \sum_{y \in X_{k(x)+1}} \bar{P}_{i(t)}(y|x, a)(\mu^-(x, a, y) - \mu^+(x, a, y))$$

We define the following value function $v$ and error function $e$ parameterized by $\mu$, $c$ and $\beta$, and an estimated Bellman error.

$$v^\mu(x, a, x') = \mu^-(x, a, x') - \mu^+(x, a, x')$$
$$e^{\mu, \beta, c}(x, a, x') = (\mu^+(x, a, x') + \mu^-(x, a, x'))\epsilon_{i(t)}(x, a) + \beta(x') - \beta(x) + c(x)$$
$$B_t^{v, e}(x, a, x') = e(x, a, x') + v(x, a, x') - \eta \hat{\ell}_t(x, a, x') - \sum_{y \in X_{k(x)+1}} \bar{P}_{i(t)}(y|x, a)v(x, a, y)$$

So the derivative becomes

$$\frac{\partial \mathcal{L}}{\partial q(x, a, x')} = \ln \frac{q(x, a, x')}{\tilde{q}_{t+1}(x, a, x')} + \lambda_k - e^{\mu, \beta, c}(x, a, x') - v^\mu(x, a, x') + \sum_{y \in X_{k(x)+1}} \bar{P}_{i(t)}(y|x, a)v^\mu(x, a, y)$$

$$= \ln q(x, a, x') - \ln \tilde{q}_{t+1}(x, a, x') + \lambda_k - \eta \hat{\ell}_t(x, a, x') - B_t^{v^\mu, e^{\mu, \beta, c}}(x, a, x')$$

Setting the gradient to zero and using the explicit form of $\tilde{q}_{t+1}(x, a, x')$ we obtain

$$q_{t+1}(x, a, x') = \tilde{q}_{t+1}(x, a, x')e^{-\lambda_k + \eta \hat{\ell}_t(x, a, x') + B_t^{v^\mu, e^{\mu, \beta, c}}(x, a, x')}$$

$$= q_t(x, a, x')e^{-\eta \hat{\ell}_t(x, a, x')}e^{-\lambda_k + \eta \hat{\ell}_t(x, a, x') + B_t^{v^\mu, e^{\mu, \beta, c}}(x, a, x')}$$

$$= q_t(x, a, x')e^{-\lambda_k + B_t^{v^\mu, e^{\mu, \beta, c}}(x, a, x')}$$

We can use the first constraint to discover that $\lambda_k$ is a normalizer for every $k = 0, \dots, L - 1$, i.e.

$$1 = \sum_{x \in X_k} \sum_{a \in A} \sum_{x' \in X_{k+1}} q_{t+1}(x, a, x')$$

$$1 = \sum_{x \in X_k} \sum_{a \in A} \sum_{x' \in X_{k+1}} q_t(x, a, x')e^{-\lambda_k + B_t^{v^\mu, e^{\mu, \beta, c}}(x, a, x')}$$

$$e^{\lambda_k} = \sum_{x \in X_k} \sum_{a \in A} \sum_{x' \in X_{k+1}} q_t(x, a, x')e^{B_t^{v^\mu, e^{\mu, \beta, c}}(x, a, x')}$$

so defining $Z_t^k(v, e) = \sum_{x \in X_k} \sum_{a \in A} \sum_{x' \in X_{k+1}} q_t(x, a, x')e^{B_t^{v, e}(x, a, x')}$ , we obtain

$$q_{t+1}(x, a, x') = \frac{q_t(x, a, x')e^{B_t^{v^\mu, e^{\mu, \beta, c}}(x, a, x')}}{Z_t^{k(x)}(v^\mu, e^{\mu, \beta, c})} \tag{3}$$

Now to find $\beta$, $c$ and $\mu$ we consider the dual problem. Substituting $q_{t+1}$ back into $\mathcal{L}$ we obtain the following dual problem.

$$\max_{\beta, \mu \geq 0, c \geq 0} \min_q \mathcal{L}(q) = \max_{\beta, \mu \geq 0, c \geq 0} \mathcal{L}(q_{t+1}) = \max_{\beta, \mu \geq 0, c \geq 0} -\sum_{k=0}^{L-1} \ln Z_t^k(v^\mu, e^{\mu, \beta, c}) + \alpha \sum_x c(x) - 1 + \sum_{x, a, x'} \tilde{q}_{t+1}(x, a, x')$$

So after ignoring constants we observe that

$$\beta_t, \mu_t, c_t = \arg\min_{\beta, \mu \geq 0, c \geq 0} \sum_{k=0}^{L-1} \ln Z_t^k(v^\mu, e^{\mu, \beta, c}) - \alpha \sum_{x \in X} c(x) \qquad (4)$$

This a convex optimization problem with only non-negativity constraints (and no constraints about the relations between the variables), and therefore can be solved efficiently using iterative methods.

Thus we have shown that step (2) can be implemented efficiently by solving (4) which is a convex optimization problem with only non-negativity constraints, and then using the result to compute $q_{t+1}$ as described in (3).

# B   pseudo-code for "Bounded Bandit UC-O-REPS"

---

**Algorithm 1** Bounded Bandit UC-O-REPS Algorithm

---

**Parameters:** state space $X$, action space $A$, time horizon $T$, minimum reachability parameter $\alpha$, optimization parameter $\eta$ and confidence parameter $\delta$.

**Initialization:**
start first epoch: $i(1) \leftarrow 1$
initialize visit counters $\forall (x, a, x') \in X \times A \times X_{k(x)+1}$: $N_1(x, a) \leftarrow 0, N_1(x, a, x') \leftarrow 0$
initialize in-epoch visit counters $\forall (x, a, x') \in X \times A \times X_{k(x)+1}$: $n_1(x, a) \leftarrow 0, n_1(x, a, x') \leftarrow 0$
initialize first policy $\forall (x, a) \in X \times A$: $\pi_1(a|x) \leftarrow \frac{1}{|A|}$
initialize first occupancy measure $\forall (x, a, x') \in X \times A \times X_{k(x)+1}$: $q_1(x, a, x') \leftarrow \frac{1}{|X_k||A||X_{k+1}|}$

**for** $t = 1$ **to** $T$ **do**
    traverse trajectory $U^{(t)} = (x_0^{(t)}, a_0^{(t)}, \ldots, x_{L-1}^{(t)}, a_{L-1}^{(t)}, x_L^{(t)})$ using policy $\pi_t$
    observe losses $\ell_t(U^{(t)}) = \left\{\ell_t(x_k^{(t)}, a_k^{(t)})\right\}_{k=0}^{L-1}$
    update in-epoch counters $\forall k = 0, \ldots, L-1$:

$$n_{i(t)}(x_k^{(t)}, a_k^{(t)}) \leftarrow n_{i(t)}(x_k^{(t)}, a_k^{(t)}) + 1$$
$$n_{i(t)}(x_k^{(t)}, a_k^{(t)}, x_{k+1}^{(t)}) \leftarrow n_{i(t)}(x_k^{(t)}, a_k^{(t)}, x_{k+1}^{(t)}) + 1$$

  **if** $\exists (x, a) \in X \times A.\ \ n_{i(t)}(x, a) \geq N_{i(t)}(x, a)$ **then**
    start new epoch: $i(t+1) \leftarrow i(t) + 1$
    initialize epoch counters $\forall (x, a, x') \in X \times A \times X_{k(x)+1}$: $n_{i(t+1)}(x, a) \leftarrow 0, n_{i(t+1)}(x, a, x') \leftarrow 0$
    update total counters $\forall (x, a, x') \in X \times A \times X_{k(x)+1}$:

$$N_{i(t+1)}(x, a) \leftarrow N_{i(t)}(x, a) + n_{i(t)}(x, a)$$
$$N_{i(t+1)}(x, a, x') \leftarrow N_{i(t)}(x, a, x') + n_{i(t)}(x, a, x')$$

    compute transition estimate $\forall (x, a, x') \in X \times A \times X_{k(x)+1}$: $\bar{P}_{i(t+1)}(x'|x, a) \leftarrow \frac{N_{i(t+1)}(x, a, x')}{\max\{1, N_{i(t+1)}(x, a)\}}$
  **else**
    continue in the same epoch: $i(t+1) \leftarrow i(t)$
  compute policy for next episode: $q_{t+1}, \pi_{t+1} \leftarrow \texttt{Comp-Policy}(X, A, T, \alpha, \eta, \delta, q_t, \bar{P}_{i(t+1)}, N_{i(t+1)}, \ell_t(U^{(t)}))$

---

---

**Algorithm 2** Comp-Policy Procedure

---

**Input:** state space $X$, action space $A$, time horizon $T$, minimum reachability parameter $\alpha$, optimization parameter $\eta$ and confidence parameter $\delta$, previous occupancy measure $q_t$, transition function estimate $\bar{P}_{i(t)}$, visit counters $N_{i(t)}$ and obtained losses $\ell_t(U^{(t)})$.

compute loss function estimate:

$$\hat{\ell}_t(x,a) = \begin{cases} \frac{\ell_t(x,a)}{q_t(x,a)}, & \text{if } (x,a) \in U^{(t)} \\ 0, & \text{otherwise} \end{cases}$$

compute confidence set size parameter:

$$\epsilon_{i(t)}(x,a) = \sqrt{\frac{2|X_{k(x)+1}| \ln \frac{T|X||A|}{\delta}}{\max\{1, N_{i(t)}(x,a)\}}}$$

define functions:

$$v^{\mu}(x,a,x') = \mu^-(x,a,x') - \mu^+(x,a,x')$$
$$e^{\mu,\beta,c}(x,a,x') = (\mu^+(x,a,x') + \mu^-(x,a,x'))\epsilon_{i(t)}(x,a) + \beta(x') - \beta(x) + c(x)$$
$$B_t^{v,e}(x,a,x') = e(x,a,x') + v(x,a,x') - \eta\hat{\ell}_t(x,a,x') - \sum_{y \in X_{k(x)+1}} \bar{P}_{i(t)}(y|x,a)v(x,a,y)$$
$$Z_t^k(v,e) = \sum_{x \in X_k} \sum_{a \in A} \sum_{x' \in X_{k+1}} q_t(x,a,x')e^{B_t^{v,e}(x,a,x')}$$

solve optimization problem:

$$\beta_t, \mu_t, c_t = \arg \min_{\beta,\mu \geq 0, c \geq 0} \sum_{k=0}^{L-1} \ln Z_t^k(v^{\mu}, e^{\mu,\beta,c}) - \alpha \sum_{x \in X} c(x)$$

compute next occupancy measure $\forall (x,a,x') \in X \times A \times X_{k(x)+1}$:

$$q_{t+1}(x,a,x') = \frac{q_t(x,a,x')e^{B_t^{v^{\mu_t},e^{\mu_t,\beta_t,c_t}}(x,a,x')}}{Z_t^{k(x)}(v^{\mu_t}, e^{\mu_t,\beta_t,c_t})}$$

compute next policy $\forall (x,a) \in X \times A$:

$$\pi_{t+1}(a|x) = \frac{\sum_{x' \in X_{k(x)+1}} q_{t+1}(x,a,x')}{\sum_{b \in A} \sum_{x' \in X_{k(x)+1}} q_{t+1}(x,b,x')}$$

---

# C  pseudo-code for "Shifted Bandit UC-O-REPS"

---

**Algorithm 3** Shifted Bandit UC-O-REPS Algorithm

---

**Parameters:** state space $X$, action space $A$, time horizon $T$, perturbation parameter $\rho$, optimization parameter $\eta$ and confidence parameter $\delta$.

**Initialization:**
start first epoch: $i(1) \leftarrow 1$
initialize visit counters $\forall (x, a, x') \in X \times A \times X_{k(x)+1}$: $N_1(x, a) \leftarrow 0, N_1(x, a, x') \leftarrow 0$
initialize in-epoch visit counters $\forall (x, a, x') \in X \times A \times X_{k(x)+1}$: $n_1(x, a) \leftarrow 0, n_1(x, a, x') \leftarrow 0$
initialize first policy $\forall (x, a) \in X \times A$: $\pi_1(a|x) \leftarrow \frac{1}{|A|}$
initialize first occupancy measure $\forall (x, a, x') \in X \times A \times X_{k(x)+1}$: $q_1(x, a, x') \leftarrow \frac{1}{|X_k||A||X_{k+1}|}$

**for** $t = 1$ **to** $T$ **do**
    traverse trajectory $U^{(t)} = (x_0^{(t)}, a_0^{(t)}, \ldots, x_{L-1}^{(t)}, a_{L-1}^{(t)}, x_L^{(t)})$ using policy $\pi_t$
    observe losses $\ell_t(U^{(t)}) = \left\{ \ell_t(x_k^{(t)}, a_k^{(t)}) \right\}_{k=0}^{L-1}$
    update in-epoch counters $\forall k = 0, \ldots, L-1$:

$$n_{i(t)}(x_k^{(t)}, a_k^{(t)}) \leftarrow n_{i(t)}(x_k^{(t)}, a_k^{(t)}) + 1$$
$$n_{i(t)}(x_k^{(t)}, a_k^{(t)}, x_{k+1}^{(t)}) \leftarrow n_{i(t)}(x_k^{(t)}, a_k^{(t)}, x_{k+1}^{(t)}) + 1$$

    **if** $\exists (x, a) \in X \times A. \quad n_{i(t)}(x, a) \geq N_{i(t)}(x, a)$ **then**
        start new epoch: $i(t+1) \leftarrow i(t) + 1$
        initialize epoch counters $\forall (x, a, x') \in X \times A \times X_{k(x)+1}$: $n_{i(t+1)}(x, a) \leftarrow 0, n_{i(t+1)}(x, a, x') \leftarrow 0$
        update total counters $\forall (x, a, x') \in X \times A \times X_{k(x)+1}$:

$$N_{i(t+1)}(x, a) \leftarrow N_{i(t)}(x, a) + n_{i(t)}(x, a)$$
$$N_{i(t+1)}(x, a, x') \leftarrow N_{i(t)}(x, a, x') + n_{i(t)}(x, a, x')$$

        compute transition estimate $\forall (x, a, x') \in X \times A \times X_{k(x)+1}$: $\bar{P}_{i(t+1)}(x'|x, a) \leftarrow \frac{N_{i(t+1)}(x, a, x')}{\max\{1, N_{i(t+1)}(x, a)\}}$
        compute perturbed transition estimate $\forall (x, a, x') \in X \times A \times X_{k(x)+1}$:

$$\bar{P}_{i(t+1)}^{\star}(x'|x, a) \leftarrow (1-\rho)\bar{P}_{i(t+1)}(x'|x, a) + \frac{\rho}{|X_{k(x)+1}|}$$

    **else**
        continue in the same epoch: $i(t+1) \leftarrow i(t)$
    compute next episode policy: $q_{t+1}, \pi_{t+1} \leftarrow \texttt{Comp-Policy}(X, A, T, \frac{\rho}{|X|}, \eta, \delta, q_t, \bar{P}_{i(t+1)}^{\star}, N_{i(t+1)}, \ell_t(U^{(t)}))$

---

# D    general claims about occupancy measures

**Lemma D.1.** *Let $P$ be a transition function and let $\pi,\tilde{\pi}$ be policies such that $\|\pi(\cdot|x) - \tilde{\pi}(\cdot|x)\|_1 \leq \nu$ for every $x$. Then, the following equations hold,*

$$\sum_{x\in X} |q^{P,\pi}(x) - q^{P,\tilde{\pi}}(x)| \leq \sum_{x\in X}\sum_{a\in A} |q^{P,\pi}(x,a) - q^{P,\tilde{\pi}}(x,a)| \tag{5}$$

$$\sum_{x\in X}\sum_{a\in A} |q^{P,\pi}(x,a) - q^{P,\tilde{\pi}}(x,a)| = \sum_{x\in X}\sum_{a\in A}\sum_{x'\in X_{k(x)+1}} |q^{P,\pi}(x,a,x') - q^{P,\tilde{\pi}}(x,a,x')| \tag{6}$$

$$\sum_{x\in X}\sum_{a\in A}\sum_{x'\in X_{k(x)+1}} |q^{P,\pi}(x,a,x') - q^{P,\tilde{\pi}}(x,a,x')| \leq \sum_{x\in X} |q^{P,\pi}(x) - q^{P,\tilde{\pi}}(x)| + L\nu \tag{7}$$

*Proof.* First of all notice that,

$$\sum_x |q^{P,\pi}(x) - q^{P,\tilde{\pi}}(x)| = \sum_x |\sum_a q^{P,\pi}(x,a) - q^{P,\tilde{\pi}}(x,a)| \leq \sum_{x,a} |q^{P,\pi}(x,a) - q^{P,\tilde{\pi}}(x,a)|.$$

Now notice that,

$$
\begin{aligned}
\sum_{x,a,x'} |q^{P,\pi}(x,a,x') - q^{P,\tilde{\pi}}(x,a,x')| &= \sum_{x,a,x'} |q^{P,\pi}(x,a)P(x'|x,a) - q^{P,\tilde{\pi}}(x,a)P(x'|x,a)| \\
&= \sum_{x,a,x'} P(x'|x,a)|q^{P,\pi}(x,a) - q^{P,\tilde{\pi}}(x,a)| \\
&= \sum_{x,a} |q^{P,\pi}(x,a) - q^{P,\tilde{\pi}}(x,a)|\sum_{x'} P(x'|x,a) \\
&= \sum_{x,a} |q^{P,\pi}(x,a) - q^{P,\tilde{\pi}}(x,a)|.
\end{aligned}
$$

Finally we have that,

$$
\begin{aligned}
\sum_{x,a} |q^{P,\pi}(x,a) - q^{P,\tilde{\pi}}(x,a)| &= \sum_{x,a} |q^{P,\pi}(x)\pi(a|x) - q^{P,\tilde{\pi}}(x)\tilde{\pi}(a|x)| \\
&\leq \sum_{x,a} |q^{P,\pi}(x)\pi(a|x) - q^{P,\tilde{\pi}}(x)\pi(a|x)| + |q^{P,\tilde{\pi}}(x)\pi(a|x) - q^{P,\tilde{\pi}}(x)\tilde{\pi}(a|x)| \\
&= \sum_{x,a} \pi(a|x)|q^{P,\pi}(x) - q^{P,\tilde{\pi}}(x)| + q^{P,\tilde{\pi}}(x)|\pi(a|x) - \tilde{\pi}(a|x)| \\
&= \sum_x |q^{P,\pi}(x) - q^{P,\tilde{\pi}}(x)|\sum_a \pi(a|x) + \sum_x q^{P,\tilde{\pi}}(x)\sum_a |\pi(a|x) - \tilde{\pi}(a|x)| \\
&\leq \sum_x |q^{P,\pi}(x) - q^{P,\tilde{\pi}}(x)| + \sum_x q^{P,\tilde{\pi}}(x)\nu = \sum_x |q^{P,\pi}(x) - q^{P,\tilde{\pi}}(x)| + L\nu.
\end{aligned}
$$

$\square$

**Lemma D.2.** *Let $\pi$ be a policy and let $P,\tilde{P}$ be transition functions such that $\|P(\cdot|x,a) - \tilde{P}(\cdot|x,a)\|_1 \leq \nu$*

*for every $x, a$. Then, the following equations hold,*

$$\sum_{x \in X} |q^{P,\pi}(x) - q^{\tilde{P},\pi}(x)| = \sum_{x \in X} \sum_{a \in A} |q^{P,\pi}(x,a) - q^{\tilde{P},\pi}(x,a)| \tag{8}$$

$$\sum_{x \in X} \sum_{a \in A} |q^{P,\pi}(x,a) - q^{\tilde{P},\pi}(x,a)| \leq \sum_{x \in X} \sum_{a \in A} \sum_{x' \in X_{k(x)+1}} |q^{P,\pi}(x,a,x') - q^{\tilde{P},\pi}(x,a,x')| \tag{9}$$

$$\sum_{x \in X} \sum_{a \in A} \sum_{x' \in X_{k(x)+1}} |q^{P,\pi}(x,a,x') - q^{\tilde{P},\pi}(x,a,x')| \leq \sum_{x \in X} \sum_{a \in A} |q^{P,\pi}(x,a) - q^{\tilde{P},\pi}(x,a)| + L\nu \tag{10}$$

*Proof.* First of all notice that,

$$\begin{aligned}
\sum_{x,a} |q^{P,\pi}(x,a) - q^{\tilde{P},\pi}(x,a)| &= \sum_{x,a} |q^{P,\pi}(x)\pi(a|x) - q^{\tilde{P},\pi}(x)\pi(a|x)| \\
&= \sum_{x,a} |q^{P,\pi}(x) - q^{\tilde{P},\pi}(x)|\pi(a|x) \\
&= \sum_{x} |q^{P,\pi}(x) - q^{\tilde{P},\pi}(x)| \sum_{a} \pi(a|x) \\
&= \sum_{x} |q^{P,\pi}(x) - q^{\tilde{P},\pi}(x)|.
\end{aligned}$$

Now notice that,

$$\sum_{x,a} |q^{P,\pi}(x,a) - q^{\tilde{P},\pi}(x,a)| = \sum_{x,a} |\sum_{x'} q^{P,\pi}(x,a,x') - q^{\tilde{P},\pi}(x,a,x')| \leq \sum_{x,a,x'} |q^{P,\pi}(x,a,x') - q^{\tilde{P},\pi}(x,a,x')|.$$

Finally we have that,

$$\begin{aligned}
\sum_{x,a,x'} |q^{P,\pi}(x,a,x') - q^{\tilde{P},\pi}(x,a,x')| &= \sum_{x,a,x'} |q^{P,\pi}(x,a)P(x'|x,a) - q^{\tilde{P},\pi}(x,a)\tilde{P}(x'|x,a)| \\
&\leq \sum_{x,a,x'} |q^{P,\pi}(x,a)P(x'|x,a) - q^{\tilde{P},\pi}(x,a)P(x'|x,a)| \\
&\quad + \sum_{x,a,x'} |q^{\tilde{P},\pi}(x,a)P(x'|x,a) - q^{\tilde{P},\pi}(x,a)\tilde{P}(x'|x,a)| \\
&= \sum_{x,a} |q^{P,\pi}(x,a) - q^{\tilde{P},\pi}(x,a)| \sum_{x'} P(x'|x,a) \\
&\quad + \sum_{x,a} q^{\tilde{P},\pi}(x,a) \sum_{x'} |P(x'|x,a) - \tilde{P}(x'|x,a)| \\
&\leq \sum_{x,a} |q^{P,\pi}(x,a) - q^{\tilde{P},\pi}(x,a)| + L\nu.
\end{aligned}$$

$\square$

# E  distance between occupancy measures

**Lemma E.1.** *Let $P$ be a transition function and let $\pi, \tilde{\pi}$ be policies such that $\|\pi(\cdot|x) - \tilde{\pi}(\cdot|x)\|_1 \leq \nu$ for every $x$. Then, for every $k = 0, \ldots, L-1$,*

$$\sum_{x_k \in X_k} |q^{P,\pi}(x_k) - q^{P,\tilde{\pi}}(x_k)| \leq k\nu$$

*Proof.* by induction on $k$. For $k = 0$ we have that

$$\sum_{x_0 \in X_0} |q^{P,\pi}(x_0) - q^{P,\tilde{\pi}}(x_0)| = |q^{P,\pi}(x_0) - q^{P,\tilde{\pi}}(x_0)| = |1 - 1| = 0.$$

Assume that the claim is correct for every $s < k$ and prove for $k$.

$$\sum_{x_k \in X_k} |q^{P,\pi}(x_k) - q^{P,\tilde{\pi}}(x_k)| =$$

$$= \sum_{x_k} \left| \sum_{x_{k-1}, a_{k-1}} q^{P,\pi}(x_{k-1}, a_{k-1}) P(x_k|x_{k-1}, a_{k-1}) - q^{P,\tilde{\pi}}(x_{k-1}, a_{k-1}) P(x_k|x_{k-1}, a_{k-1}) \right|$$

$$\leq \sum_{x_k} \sum_{x_{k-1}, a_{k-1}} P(x_k|x_{k-1}, a_{k-1}) |q^{P,\pi}(x_{k-1}, a_{k-1}) - q^{P,\tilde{\pi}}(x_{k-1}, a_{k-1})|$$

$$= \sum_{x_{k-1}, a_{k-1}} |q^{P,\pi}(x_{k-1}, a_{k-1}) - q^{P,\tilde{\pi}}(x_{k-1}, a_{k-1})| \sum_{x_k} P(x_k|x_{k-1}, a_{k-1})$$

$$= \sum_{x_{k-1}, a_{k-1}} |q^{P,\pi}(x_{k-1})\pi(a_{k-1}|x_{k-1}) - q^{P,\tilde{\pi}}(x_{k-1})\tilde{\pi}(a_{k-1}|x_{k-1})|$$

$$\leq \sum_{x_{k-1}, a_{k-1}} |q^{P,\pi}(x_{k-1})\pi(a_{k-1}|x_{k-1}) - q^{P,\pi}(x_{k-1})\tilde{\pi}(a_{k-1}|x_{k-1})|$$

$$+ \sum_{x_{k-1}, a_{k-1}} |q^{P,\pi}(x_{k-1})\tilde{\pi}(a_{k-1}|x_{k-1}) - q^{P,\tilde{\pi}}(x_{k-1})\tilde{\pi}(a_{k-1}|x_{k-1})|$$

$$= \sum_{x_{k-1}, a_{k-1}} q^{P,\pi}(x_{k-1}) |\pi(a_{k-1}|x_{k-1}) - \tilde{\pi}(a_{k-1}|x_{k-1})|$$

$$+ \sum_{x_{k-1}, a_{k-1}} \tilde{\pi}(a_{k-1}|x_{k-1}) |q^{P,\pi}(x_{k-1}) - q^{P,\tilde{\pi}}(x_{k-1})|$$

$$= \sum_{x_{k-1}} q^{P,\pi}(x_{k-1}) \sum_{a_{k-1}} |\pi(a_{k-1}|x_{k-1}) - \tilde{\pi}(a_{k-1}|x_{k-1})|$$

$$+ \sum_{x_{k-1}} |q^{P,\pi}(x_{k-1}) - q^{P,\tilde{\pi}}(x_{k-1})| \sum_{a_{k-1}} \tilde{\pi}(a_{k-1}|x_{k-1})$$

$$\leq \sum_{x_{k-1}} q^{P,\pi}(x_{k-1})\nu + \sum_{x_{k-1}} |q^{P,\pi}(x_{k-1}) - q^{P,\tilde{\pi}}(x_{k-1})|$$

$$\leq \nu + (k-1)\nu = k\nu$$

where the last inequality follows from the induction hypothesis. $\square$

**Corollary E.1.** *Let $P$ be a transition function and let $\pi, \tilde{\pi}$ be policies such that $\|\pi(\cdot|x) - \tilde{\pi}(\cdot|x)\|_1 \leq \nu$ for every $x$. Then,*

$$\|q^{P,\pi} - q^{P,\tilde{\pi}}\|_1 \leq O(L^2 \nu)$$

*Proof.* We use Lemma E.1 and Lemma D.1 to obtain,

$$\|q^{P,\pi} - q^{P,\tilde{\pi}}\|_1 = \sum_{x,a,x'} |q^{P,\pi}(x,a,x') - q^{p,\tilde{\pi}}(x,a,x')|$$

$$\leq \sum_{x \in X} |q^{P,\pi}(x) - q^{P,\tilde{\pi}}(x)| + L\nu$$

$$= \sum_{k=0}^{L-1} \sum_{x_k \in X_k} |q^{P,\pi}(x_k) - q^{P,\tilde{\pi}}(x_k)| + L\nu$$

$$\leq \sum_{k=0}^{L-1} k\nu + L\nu \leq L^2\nu + L\nu = O(L^2\nu).$$

$\square$

**Lemma E.2.** *Let $\pi$ be a policy and let $P, \tilde{P}$ be transition functions such that $\|P(\cdot|x,a) - \tilde{P}(\cdot|x,a)\|_1 \leq \nu$ for every $x, a$. Then, for every $k = 0, \ldots, L-1$,*

$$\sum_{x_k \in X_k} |q^{P,\pi}(x_k) - q^{\tilde{P},\pi}(x_k)| \leq k\nu$$

*Proof.* by induction on $k$. For $k = 0$ we have that

$$\sum_{x_0 \in X_0} |q^{P,\pi}(x_0) - q^{\tilde{P},\pi}(x_0)| = |q^{P,\pi}(x_0) - q^{\tilde{P},\pi}(x_0)| = |1 - 1| = 0.$$

Assume that the claim is correct for every $s < k$ and prove for $k$.

$$\sum_{x_k \in X_k} |q^{P,\pi}(x_k) - q^{\tilde{P},\pi}(x_k)| =$$

$$= \sum_{x_k} \left| \sum_{x_{k-1},a_{k-1}} q^{P,\pi}(x_{k-1},a_{k-1})P(x_k|x_{k-1},a_{k-1}) - q^{\tilde{P},\pi}(x_{k-1},a_{k-1})\tilde{P}(x_k|x_{k-1},a_{k-1}) \right|$$

$$\leq \sum_{x_{k-1},a_{k-1},x_k} |q^{P,\pi}(x_{k-1},a_{k-1})P(x_k|x_{k-1},a_{k-1}) - q^{\tilde{P},\pi}(x_{k-1},a_{k-1})\tilde{P}(x_k|x_{k-1},a_{k-1})|$$

$$\leq \sum_{x_{k-1},a_{k-1},x_k} |q^{P,\pi}(x_{k-1},a_{k-1})P(x_k|x_{k-1},a_{k-1}) - q^{P,\pi}(x_{k-1},a_{k-1})\tilde{P}(x_k|x_{k-1},a_{k-1})|$$

$$+ \sum_{x_{k-1},a_{k-1},x_k} |q^{P,\pi}(x_{k-1},a_{k-1})\tilde{P}(x_k|x_{k-1},a_{k-1}) - q^{\tilde{P},\pi}(x_{k-1},a_{k-1})\tilde{P}(x_k|x_{k-1},a_{k-1})|$$

$$= \sum_{x_{k-1},a_{k-1}} q^{P,\pi}(x_{k-1},a_{k-1}) \sum_{x_k} |P(x_k|x_{k-1},a_{k-1}) - \tilde{P}(x_k|x_{k-1},a_{k-1})|$$

$$+ \sum_{x_{k-1},a_{k-1}} |q^{P,\pi}(x_{k-1},a_{k-1}) - q^{\tilde{P},\pi}(x_{k-1},a_{k-1})| \sum_{x_k} \tilde{P}(x_k|x_{k-1},a_{k-1})$$

$$\leq \nu \sum_{x_{k-1},a_{k-1}} q^{P,\pi}(x_{k-1},a_{k-1}) + \sum_{x_{k-1},a_{k-1}} |q^{P,\pi}(x_{k-1},a_{k-1}) - q^{\tilde{P},\pi}(x_{k-1},a_{k-1})|$$

$$= \nu + \sum_{x_{k-1}} |q^{P,\pi}(x_{k-1}) - q^{\tilde{P},\pi}(x_{k-1})| \leq \nu + (k-1)\nu = k\nu$$

where the last inequality follows from the induction hypothesis. $\square$

**Corollary E.2.** *Let $\pi$ be a policy and let $P,\tilde{P}$ be transition functions such that $\|P(\cdot|x,a) - \tilde{P}(\cdot|x,a)\|_1 \leq \nu$ for every $x,a$. Then,*

$$\|q^{P,\pi} - q^{\tilde{P},\pi}\|_1 \leq O(L^2\nu)$$

*Proof.* We use Lemma E.2 and Lemma D.2 to obtain,

$$\begin{aligned}
\|q^{P,\pi} - q^{\tilde{P},\pi}\|_1 &= \sum_{x,a,x'} |q^{P,\pi}(x,a,x') - q^{\tilde{P},\pi}(x,a,x')| \\
&\leq \sum_{x\in X} |q^{P,\pi}(x) - q^{\tilde{P},\pi}(x)| + L\nu \\
&= \sum_{k=0}^{L-1} \sum_{x_k \in X_k} |q^{P,\pi}(x_k) - q^{\tilde{P},\pi}(x_k)| + L\nu \\
&\leq \sum_{k=0}^{L-1} k\nu + L\nu \leq L^2\nu + L\nu = O(L^2\nu).
\end{aligned}$$

$\square$

# F    proof of Lemma 6.3

Since the original UC-O-REPS algorithm is performed with respect to the sequence of loss functions $\{\hat{\ell}_t\}_{t=1}^T$ and the set of occupancy measures $\Delta_\alpha(M)$, we can use the regret analysis of UC-O-REPS to obtain the following with probability of at least $1-\delta$ (with probability of at most $\delta$ we can bound it as $\frac{TL}{\alpha}$ and then setting $\delta = \frac{|X||A|}{T}$ eliminates this term),

$$\mathbb{E}\left[\sum_{t=1}^T \langle q^{P_t,\pi_t} - q, \hat{\ell}_t \rangle\right] \leq \mathbb{E}\left[\sum_{t=1}^T \langle q_t - \tilde{q}_{t+1}, \hat{\ell}_t \rangle\right] + \frac{D(q\|q_1)}{\eta}. \tag{11}$$

Using the exact form of $\tilde{q}_{t+1}$ and the fact that $e^x \geq 1 + x$, we get that

$$\tilde{q}_{t+1}(x,a) \geq q_t(x,a) - \eta q_t(x,a)\hat{\ell}_t(x,a)$$

and therefore

$$\begin{aligned}
\mathbb{E}\left[\sum_{t=1}^T \langle q_t - \tilde{q}_{t+1}, \hat{\ell}_t \rangle\right] &\leq \eta\mathbb{E}\left[\sum_{t=1}^T \sum_{x,a} q_t(x,a)\hat{\ell}_t^2(x,a)\right] \\
&\leq \eta\mathbb{E}\left[\sum_{t=1}^T \sum_{x,a} q_t(x,a)\frac{\ell_t(x,a)}{q_t(x,a)}\hat{\ell}_t(x,a)\right] \\
&\leq \eta\mathbb{E}\left[\sum_{t=1}^T \sum_{x,a} \hat{\ell}_t(x,a)\right] \\
&= \eta\mathbb{E}\left[\sum_{t=1}^T \mathbb{E}\left[\sum_{x,a} \hat{\ell}_t(x,a)\Big|U^{(1)},\ldots,U^{(t-1)}\right]\right] \\
&= \eta\mathbb{E}\left[\sum_{t=1}^T \sum_{x,a} q^{P,\pi_t}(x)\frac{\ell_t(x,a)}{q^{P_t,\pi_t}(x)}\right] \leq \frac{\eta}{\alpha}\mathbb{E}\left[\sum_{t=1}^T \sum_{x,a} q^{P,\pi_t}(x)\right] = \frac{\eta L|A|T}{\alpha}
\end{aligned}$$

where in the last inequality we used the fact that $q^{P_t,\pi_t} \in \Delta_\alpha(M,t)$, and therefore $q^{P_t,\pi_t}(x) \geq \alpha$ for every $x$. For the second term, $D(q_\alpha \| q_1)/\eta$, we use the fact that the unnormalized KL divergence is the Bregman divergence associated with the unnormalized negative entropy defined as follows,

$$R(q) = \sum_{x,a} q(x,a) \ln q(x,a) - q(x,a).$$

Now from standard arguments we obtain

$$D(q\|q_1) \leq R(q) - R(q_1) \leq \sum_{x \in X} \sum_{a \in A} q_1(x,a) \ln \frac{1}{q_1(x,a)} \leq \sum_{k=0}^{L-1} \ln |X_k||A| \leq L \ln \frac{|X||A|}{L}.$$

Putting these two bounds together completes the proof.