[Reviews · NeurIPS 2019]

Reviewer 1



The submission studies the adversarial online learning in episodic loop-free Markov decision processes. The importance of this work is that it is the first to provide the understanding to an adversarial online learning problem where the transition function is unknown, the loss functions are changing, and each feedback is bandit. The related work clearly describe the line of this research field from fixing an unknown transition and an unknown loss function to the setting studied in this submission. Although the MDPs considered in the submission is L-layered and loop-free, the results and the analysis pave the way for general MDPs. The main idea is the design of the confidence sets to include the optimal occupancy measure which induces the optimal policy. Two ways to construct the confidence sets are proposed, one with the reachable probability \beta assumption and the other without. The confidence sets are also responsible for producing policies with sufficient degree of exploration. The other idea is the construction of estimated loss functions from bandit feedback, so that an optimistic occupancy measure balancing the trade-off between the loss and the distance to the previous chosen occupancy measure is computable. Two algorithms are proposed and analyzed. When the MDP satisfies the \beta > 0 assumption, Bounded Bandit UC-O-REPS achieves an O ̃(L|X|\sqrt{|A|T}/^beta) regret. When the \beta > 0 assumption is removed, Shitted Bandit UC-O-REPS achieves a regret bound of O ̃(L^{3/2}|X||A|^{¼} T^{ ¾} ). Originality. The submission is the first one to tackle the two challenges, unknown transition and bandit feedback, at the same time. Quality. Rigorous and clear-to-follow proofs of the analysis are provided in the supplementary. Clarity. The paper and the analysis are self content and well written. Significance. The submission provides insights and solutions to the bandit online MDP learning. Although the problem instances are restricted to episodic loop-free MDPs, the submission pave the way for general MDPs. ------------------------------ Update Thank you for the response. The feedback from the authors is appreciated has been reflected in the overall score for the submission.

Reviewer 2



After reading the response, I believe the authors could make the problem formulation and algorithms more clear. I am raising my score accordingly. ================================================== Overall, I feel this paper is not self-contained and incomplete. There is no complete algorithm being presented. Some of the notations are not well-defined and somewhat confusing. The authors also spend much effort explaining the proofs while the problem formulations and algorithms are not clear. I believe the priorities should be reversed. My detailed comments are as follows: 1. Section 4 on page 4: This seems to be a partial/incomplete introduction to UCRL type algorithms. Many of the key components in the UCRL algorithm which are likely needed in this paper (such as the empirical transition counts, the notion of epochs, etc.) are not explained. It is therefore not clear what is the purpose of this section. In particular, (1) the notation $\overline{P}_t$ seems being only verbally defined as the empirical transition function''. Its precise mathematical meaning is unclear. (2) The elements in $\Delta(M,t)$ seem to be probability measures q(x,a,x'). How does it also contain conditional probability measures P(x'|x,a)? It might better to define explicitly what \Delta(M,t) is. (3) What is $\epsilon_t(s,a)$? It is not defined anywhere in the paper. What is the role played by this constant and how does it affects the performance of the algorithm? 2. Section 5.1: This section is not self-contained and confusing as there is no algorithm presented. Instead, the authors simply refer to a previous paper [4] and mention a few things that need to be changed there. I believe this is not a good way of presenting algorithms as the readers have to look back into [4] in detail and connect dots and pieces in order to understand what the algorithm does. 3. Section 5.2: Since UC-O-REPS is not clearly explained and the empirical transitions are not defined. It is rather difficult to follow this section, and it is not clear what is the motivation for defining a new transition $P_t^*$. 4. The proofs seem to follow the proof for UC-O-REPS [4] rather directly. However, many of the key components in UC-O-REPS are not explained. For example, this proof does not explain why the algorithm requires the notion of epochs here or why the empirical transition matrix should be updated per epoch in the way suggested by [4]. Other comments: 1. Line 130: Change different than'' to different from''. 2. First equation after Line 130: A summation is missing in the computation of expectation. 3. Line 132: For this purpose we constraint the confidence...'': Change constraint'' to constrain''. 4. Line 134: It is not clear what the elements in $\Delta_\alpha(M,t)$ are. The elements in $\Delta(M,t)$ seem to be q(x,a,x'), which are incompatible with q(x). 5. If $\Delta_\alpha(M; t) = \emptyset$, then, [$q_{t+1}$ is chosen to be an arbitrary occupancy measure].

Reviewer 3



1. Originality and Quality. This is work is original to the best of my knowledge. Adequate citation is given to related work. 2. Clarity. The paper is very well written and straight-forward. Adequate intuition and discussion are provided in the paper. The writing is reader-friendly as proofs for important lemmas are included in the main paper. 3. Significance. The idea of the first algorithm is straight-forward but the proof needs slight modification due to the bias. The idea of the second algorithm is not very trivial. ========= Update ========= Thank you for the feedback. My overall score remains the same.

[Author Response · NeurIPS 2019]

We would like to thank the reviewers for their important thorough reviews. We are happy to make use of their insightful comments in order to make this paper as clear and as complete as possible.

First of all, we would like to point out some important comments made by the reviewers, that will also be added to the final discussion of the paper. This paper presents the first no-regret algorithms for adversarial online MDPs with both an unknown transition function and a bandit feedback. It can and should pave the way for future algorithms to remove the loop-free assumption and to achieve tighter regret bounds. Handling both of these factors makes this problem a very hard one, as can be seen by the fact that it remained completely untouched while all the other variations of the online MDP setting were pretty much solved.

This setting combines two challenges that together become extra difficult, while previous papers only have to face one of them. When the transition function is known but feedback is bandit, this is actually an online linear optimization problem so importance sampling can be implemented without a problem. When the transition function is unknown but feedback is full information, the problem is more difficult but the learner does not need to estimate the loss, so she just needs to follow the regularized leader while estimating the transition function and building confidence sets around it.

When the two challenges are combined, this gives rise to new problems. First, the learner cannot use an unbiased estimator anymore and therefore the bias must be handled meticulously to make sure that it does not sum up to a large drift. Second, the learner cannot just follow the regularized leader because she has no idea what is going on in parts of the MDP that she did not visit enough, and the adversary can take advantage of the fact that the learner must explore more than in other scenarios. This is where the $\beta > 0$ assumption becomes so important, and getting rid of it is the most difficult part. The technique we use (perturbed transition function) and the way to analyze it are entirely novel, and it could pave the way for more advanced techniques that will not suffer the extra $O(T^{1/4})$ regret.

**Reviewer** #1**:** A discussion with all the comments made above about hardness and comparison to [4] will be added.

**Reviewer** #2**:** We would like to mention that the problem formulation is fully described in Section 2 and that there are exact explanations of our algorithms. We believe that most of the confusion stems from the lack of pseudo-code (which will be added) and from the partial introduction of the UC-O-REPS algorithm in Section 4. We will make Section 4 a lot more comprehensive and will lay the foundation to make sure that the algorithms that follow it are clear.

In section 4 the steps of the algorithm are presented in the equations after line 114, and the constraints that define the confidence sets are formally defined in the supplementary material. The part that may be incomplete is maintenance of the confidence sets, but we must emphasize that these are standard techniques. Nevertheless, for completeness, we will present the counters $N_t(x, a)$ and $N_t(x, a, x')$ that the UC-O-REPS algorithm uses to count number of visits up to time $t$. We will explain that the algorithm proceeds in epochs that their purpose is to shrink the confidence sets as much as possible while they still contain $\Delta(M)$ with high probability. The confidence sets are updated in the beginning of every epoch and an epoch ends once the number of visits to some state-action pair is doubled. Then we will give the definitions of the empirical transition function (based on the counters) and of the confidence sets $\Delta(M, t)$. We will explain that $\Delta(M, t)$ is a set of occupancy measures such that their induced transition function (as described in Section 3) has $L_1$-distance of at most $\epsilon_t$ from the empirical transition function, and present $\epsilon_t(x, a)$ as a parameter that controls the size of the confidence set and scales as $\tilde{O}\left(\sqrt{|X|/N_t(x,a)}\right)$. We will mention that $\Delta(M, t)$ is constructed using constraints that are linear with respect to the occupancy measure and refer to the supplementary material where there is already a full description of the constraints and there will be pseudo-code of UC-O-REPS for completeness.

In section 5.1 we explain the changes from UC-O-REPS but we also present the steps of the algorithm in the equations after line 135. The confidence sets are maintained as before (explained in Section 4) and the constraints are found in the supplementary material. We will give the full pseudo-code of the algorithm to avoid any confusion. We will also make it clear that the new constraint of $q(x) \geq 0$ is a linear constraint because $q(x) = \sum_{a,x'} q(x, a, x')$ (as described in Section 3). We will then refer to the supplementary material where a full description of the constraints is already given.

Section 5.2 will become a lot clearer once the UC-O-REPS algorithm is adequately explained in Section 4. A discussion about the motivation for defining the perturbed transition function will also be added. The reason for this perturbation is to enforce that the learner performs sufficient exploration, since now it is not guaranteed by the $\beta > 0$ assumption. The technical reason is that it enables us to use the Bounded Bandit UC-O-REPS algorithm (it requires the $\beta$ assumption).

We would also like to point out that there is no summation missing in the computation of expectation in equation (1), in contrast to the comment of reviewer #2. The probability that a state-action pair $(x, a)$ was visited in episode $t$ is exactly $q^{P, \pi_t}(x, a)$. Finally, all the technical comments made by reviewer #2 will be fixed, and a discussion about the contribution of epochs and confidence sets to the regret analysis (which remains the same as [4]) will be added.

**Reviewer** #3**:** Previous techniques can be found in many of the papers presented in the Related Work but the perturbed transition function technique is novel.

[Meta-Review · NeurIPS 2019]

The submission considers episodic online learning of MDPs with unknown transition function and bandit feedback. A no-regret algorithm is provided in the adversarial model where the loss function for each episode can be arbitrary. However, the result is limited to the special case where the MDP is loop-free. The algorithm is based on a previous algorithm for unknown transition function but full-information feedback. The problem is important and challenging, so the loop-free case is a reasonable first step. The author feedback clarified some issues the reviewers had in particular regarding the presentation. After discussion, the reviewers all vote for accepting the submission, although their opinions are not very strong in light of the limited contribution and some weakness in the presentation.